# Some Local Fractional Inequalities Involving Fractal Sets via Generalized Exponential (*s*, *m*)-Convexity

**Wedad Saleh** [1,*] **and Adem Kılıçman** [2]

1  Department of Mathematics, Taibah University, Al-Medina 42353, Saudi Arabia
2  Department of Mathematics and Statistics, Universiti Putra Malaysia, Serdang 43400, Malaysia
*  Correspondence: wlehabi@taibahu.edu.sa

**Abstract:** Research in this paper aims to explore the concept of generalized exponentially (*s*, *m*)-convex functions, and to determine some properties of these functions. In addition, we look at some interactions between generalized exponentially (*s*, *m*)-convex functions and local fractional integrals. The properties of the generalized new special cases of (*s*, *m*)-convex functions, *s*-convex functions, and also generalized *m*-convex functions are impressive. We derive some inequalities of Hadamard's type for generalized exponentially (*s*, *m*)-convex functions, and give applications in probability density functions and generalized special methods to attest to the applicability and efficiency of the method under consideration.

**Keywords:** convex function; exponentially convex; *m*-convex function; *s*-convex function

**MSC:** 52A41; 26A51; 26D15; 26D10

## 1. Introduction and Preliminaries

In mathematics and economics, convexity is a very important property. Many researchers have developed new generalizations of convexity and also established several properties in new generalized cases. For instance, *s*-convex functions of the first type were introduced in [1] by Orlicz and the second type of *s*-convexity was introduced by Breckner in [2]; then, in [3], Hudzik and Maligranda addressed some properties of these types of *s*-convexity ($s \in (0, \infty]$).

**Definition 1.** *Assume that* $s \in (0, 1]$. *A function* $\varrho : [0, \infty) \longrightarrow \mathbb{R}$ *is called s-convex in the second sense if*

$$\varrho(t\iota_1 + (1-t)\iota_2) \le t^s \varrho(\iota_1) + (1-t)^s \varrho(\iota_2),$$

*holds for all* $\iota_1, \iota_2 \in [0, \infty)$ *and* $t \in [0, 1]$.

The next definition of *m*-convex function is given by Toader in [4].

**Definition 2.** *A function* $\varrho : [0, \iota] \longrightarrow \mathbb{R}, \iota > 0$ *is called m-convex if*

$$\varrho(t\iota_1 + m(1-t)\iota_2) \le t\varrho(\iota_1) + m(1-t)\varrho(\iota_2)$$

*holds for all* $\iota_1, \iota_1 \in [0, \iota], m \in (0, 1]$ *and* $t \in [0, 1]$.

The following definition of an exponentially convex function is given by Awan et al. [5].

**Definition 3.** *A function* $\varrho : \mu \longrightarrow \mathbb{R}$, *where* $\mu$ *is an interval, is called an exponentially convex function if*

$$\varrho(t\iota_1 + (1-t)\iota_2) \le t \frac{\varrho(\iota_1)}{\exp^{\beta\iota_1}} + (1-t) \frac{\varrho(\iota_2)}{\exp^{\beta\iota_2}},$$

*holds for all* $\iota_1, \iota_2 \in \mu, t \in [0,1]$ *and* $\beta \in \mathbb{R}$.

In the next definition, an exponentially *s*-convex function is given by Mehreen and Anwar in [6].

**Definition 4.** *A function* $\varrho : \mu \longrightarrow \mathbb{R}$, *where* $\mu$ *is an interval, is called an exponentially s-convex function if*

$$\varrho(t\iota_1 + (1-t)\iota_2) \leq t^s \frac{\varrho(\iota_1)}{\exp^{\beta \iota_1}} + (1-t)^s \frac{\varrho(\iota_2)}{\exp^{\beta \iota_2}},$$

*holds for all* $\iota_1, \iota_2 \in \mu, t \in [0,1], s \in (0,1]$ *and* $\beta \in \mathbb{R}$.

In the next definition, the $(s,m)$-convex function is given by Anastassiou in [7].

**Definition 5.** *A function* $\varrho : [0, \iota] \longrightarrow \mathbb{R}$ *is called an (s,m)-convex function if*

$$\varrho(t\iota_1 + m(1-t)\iota_2) \leq t^s \varrho(\iota_1) + m(1-t)^s \varrho(\iota_2),$$

*holds for all* $\iota_1, \iota_2 \in [0, \iota]$, $(s,m) \in (0,1] \times (0,1]$ *and* $t \in [0,1]$.

The following definition of an exponentially $(s,m)$ convex function is given by Qiang et al. in [8].

**Definition 6.** *A function* $\varrho : \mu \longrightarrow \mathbb{R}$, *where* $\mu \subseteq [0, \infty)$, *is called an exponentially (s,m) convex function in the second sense if*

$$\varrho(t\iota_1 + m(1-t)\iota_2) \leq t^s \frac{\varrho(\iota_1)}{\exp^{\beta \iota_1}} + m(1-t)^s \frac{\varrho(\iota_2)}{\exp^{\beta \iota_2}},$$

*holds for all* $\iota_1, \iota_2 \in \mu, s, m \in (0,1], t \in [0,1]$ *and* $\beta \in \mathbb{R}$.

Scientists and engineers have been paying significant attention to fractal sets and fractal theory. For Mandelbrot, a fractal set is one in which the Hausdorff dimension extends beyond the topological dimension [9,10]. Using different approaches, a wide variety of fractional calculus methods have been developed to study the properties of functions acting on fractal space [3,11–15]. Yang in [16] systematically studied and advanced on the analysis of local fractional integral functions in fractal spaces that include the local fractional calculus and function monotonicity.

Assume that $\mathbb{R}^\alpha$ is the real numbers line in fractional space. Then, upon using the concept of Gao–Yang–Kang in [16], one will be able to explain the definitions of the local fractional derivative as well as integral, respectively. If $\iota_1^\alpha, \iota_2^\alpha$ and $\iota_3^\alpha \in \mathbb{R}^\alpha$ where $0 < \alpha \leq 1$, then

1. $\iota_1^\alpha + \iota_2^\alpha \in \mathbb{R}^\alpha, \iota_1^\alpha \iota_2^\alpha \in \mathbb{R}^\alpha,$
2. $\iota_1^\alpha + \iota_2^\alpha = \iota_2^\alpha + \iota_1^\alpha = (\iota_1 + \iota_2)^\alpha = (\iota_2 + \iota_1)^\alpha,$
3. $\iota_1^\alpha + (\iota_2^\alpha + \iota_3^\alpha) = (\iota_1^\alpha + \iota_2^\alpha) + \iota_3^\alpha,$
4. $\iota_1^\alpha \iota_2^\alpha = \iota_2^\alpha \iota_1^\alpha = (\iota_1 \iota_2)^\alpha = (\iota_2 \iota_1)^\alpha,$
5. $\iota_1^\alpha (\iota_2^\alpha \iota_3^\alpha) = (\iota_1^\alpha \iota_2^\alpha) \iota_3^\alpha,$
6. $\iota_1^\alpha (\iota_2^\alpha + \iota_3^\alpha) = (\iota_1^\alpha \iota_2^\alpha) + (\iota_1^\alpha \iota_3^\alpha),$
7. $\iota_1^\alpha + 0^\alpha = 0^\alpha + \iota_1^\alpha = \iota_1^\alpha$ and $\iota_1^\alpha.1^\alpha = 1^\alpha.\iota_1^\alpha = \iota_1^\alpha.$

Next, we provide some definitions that are relevant to the local fractional calculus on $\mathbb{R}^\alpha$ which are introduced in [15,16] such as

**Definition 7.** *A non-differentiable function* $\varrho : \mathbb{R} \longrightarrow \mathbb{R}^\alpha$ *is local fractional continuous at* $\iota_0$, *if* $\forall \varepsilon > 0 \exists \delta > 0$ *such that* $|\varrho(\iota) - \varrho(\iota_0)| < \varepsilon^\delta$ *holds* $\forall |\iota - \iota_0| < \delta$ *where* $\varepsilon, \delta \in \mathbb{R}$.

The set of all locally fractional continuous functions on $(\iota_1 - \iota_2)$ is denoted by $\mathbb{C}_\alpha[\iota_1, m\iota_2]$.

**Definition 8.** *Let $\varrho$ be a local fractional continuous on $[\iota_1, \iota_2]$. The local fractional integral function $\varrho(\chi)$ having an order of $\alpha$ is defined by*

$$\begin{aligned}
{}_{\iota_1}I_{\iota_2}^{(\alpha)}\varrho(\chi) &= (\Gamma(\alpha+1))^{-1} \int_{\iota_1}^{\iota_2} \varrho(\tau)(d\tau)^\alpha \\
&= (\Gamma(\alpha+1))^{-1} \lim_{\triangle\tau \longrightarrow 0} \sum_{n=0}^{m} \varrho(\tau_n)(\triangle\tau_n)^\alpha,
\end{aligned}$$

*with $\triangle\tau_n = \tau_{n+1} - \tau_n$ and $\triangle\tau = \max \triangle\tau_n$, $n = 1, 2, \ldots, m-1$, where $[\tau_n, \tau_{n+1}]$, $n = 0, 1, \ldots, m-1$, $\tau_0 = \iota_1 < \tau_1 < \cdots < \tau_{m-1} < \tau_m = \iota_2$ is a partition on $[\iota_1, \iota_2]$, and $\Gamma$ is the Gamma function given by*

$$\Gamma(\chi) = \int_0^\infty \rho^{\chi-1} e^{-\rho} d\rho.$$

It was Mo and Sui who defined two types of generalized $s$-convex functions on fractal sets, as stated in the next definition; see [17].

**Definition 9.** *(i)　A function $\varrho\colon \mathbb{R}_+ \longrightarrow \mathbb{R}^\alpha$ is said to be a generalized $s$-convex ($s \in (0,1]$) in the first sense, if*

$$\varrho(\gamma_1\iota_1 + \gamma_2\iota_2) \le \gamma_1^{s\alpha}\varrho(\iota_1) + \gamma_2^{s\alpha}\varrho(\iota_2), \tag{1}$$

*$\forall \iota_1, \iota_2 \in \mathbb{R}_+$; $\forall \gamma_1, \gamma_2 \ge 0$ with $\gamma_1^s + \gamma_2^s = 1$.*

*(ii)　A function $\varrho\colon \mathbb{R}_+ \longrightarrow \mathbb{R}^\alpha$ is said to be a generalized $s$-convex ($s \in (0,1]$) in the second sense if (1) holds $\forall \iota_1, \iota_2 \in \mathbb{R}_+$; $\forall \gamma_1, \gamma_2 \ge 0$ with $\gamma_1 + \gamma_2 = 1$.*

Furthermore, Mo and Sui proved, in the same paper [17], that functions which are given in Definition 9, (*ii*) are non-negative.

Note that the classical $s$-convex functions are defined in the first (second) sense, if $\alpha = 1$ in Definition 9; see [18].

The aforementioned study also established a variant of Dragomir and Fitzatrick's Hadamard's inequality holding for $s$-convex functions in the second sense.

**Theorem 1.** *Assume that $\varrho\colon \mathbb{R}_+ \longrightarrow \mathbb{R}_+$ is a $s$-convex function in the second sense, $s \in (0,1]$ and $\iota_1, \iota_2 \in \mathbb{R}_+$, $\iota_1 < \iota_2$. If $\varrho \in L^1([\iota_1, \iota_2])$, then*

$$2^{s-1}\varrho\left(\frac{\iota_1 + \iota_2}{2}\right) \le \frac{1}{\iota_2 - \iota_1} \int_{\iota_1}^{\iota_2} \varrho(x)dx \le \frac{\varrho(\iota_1) + \varrho(\iota_2)}{s+1}. \tag{2}$$

The variation of the generalized Hadamar inequality applies to the generalized $s$-convex function in the second sense in [19].

**Theorem 2.** *Assume that $\varrho\colon \mathbb{R}_+ \longrightarrow \mathbb{R}_+^\alpha$ is a generalized $s$-convex function in the second sense where $0 < s \le 1$ and $\iota_1, \iota_2 \in \mathbb{R}_+$ with $\iota_1 < \iota_2$. If $\varrho \in L^1([\iota_1, \iota_2])$, then*

$$2^{\alpha(s-1)}\varrho\left(\frac{\iota_1 + \iota_2}{2}\right) \le \frac{\Gamma(\alpha+1)}{(\iota_2 - \iota_1)^\alpha} {}_{\iota_1}I_{\iota_2}^{(\alpha)}\varrho(x) \le \frac{\Gamma(s\alpha+1)\Gamma(\alpha+1)}{\Gamma((s+1)\alpha+1)}(\varrho(\iota_1) + \varrho(\iota_2)). \tag{3}$$

Making use of local fractional integrals, some researchers have looked into various well-known integral inequalities. For example, Kilicman and Saleh [20,21] established generalized HH inequalities for generalized s-convex functions. For generalized m-convex functions on fractal sets with utilities, Du et al. [14] considered several inequalities. The generalized Jensen and HH inequalities for $h$-convex functions have also been investigated by Vivas et al. [22]. Generalized $(s, m)$-convex function-related generalized Hermite-Hadamard (HH) type inequalities have been discovered by Abdeljawad et al. [23]

Owing to the aforementioned trend and inspired by the ongoing activities, the rest of this paper is organized as follows: first of all, in Section 2, we define and explore the newly introduced idea about generalized convex functions and their algebraic properties, which is called generalized exponentially $(s, m)$-convex functions and denoted by $GE^2_{s,m}$. In addition, we establish some results regarding the interaction between generalized exponentially $(s, m)$-convex functions as well as local fractional integrals. There are some interesting properties among the generalized new special cases of $(s, m)$-convex functions, $s$-convex functions, and also generalized $m$-convex functions. In Section 3, we present the novel version of Hermite–Hadamard type inequality. Finally, we give some applications in support of the newly introduced idea and a brief conclusion.

## 2. Generalized Exponentially $(s, m)$-Convex Functions and Associated Algebraic Properties

We now add and introduce a new concept of a new family of convex functions that is called generalized exponential $(s, m)$-convex functions on fractal space, and explore some of their properties.

**Definition 10.** *Assume that $s \in (0, 1]$ and $\mu \subseteq [0, \infty)$ is an interval. A function $\varrho : \mu \longrightarrow \mathbb{R}^\alpha$ is called generalized exponentially $(s, m)$-convex function in the second sense if*

$$\varrho(t\iota_1 + m(1 - t)\iota_2) \leq t^{\alpha s} \frac{\varrho(\iota_1)}{\exp^{\theta\iota_1}} + m^\alpha (1 - t)^{\alpha s} \frac{\varrho(\iota_2)}{\exp^{\theta\iota_2}},$$

*holds for all $\iota_1, \iota_1 \in \mu$, $m \in (0, 1]$ and $\theta \in \mathbb{R}$, denoted by $GE^2_{s,m}$.*

**Example 1.** *Assume that a function $\varrho : [0, \infty) \longrightarrow \mathbb{R}^\alpha$ is defined by $\varrho = \ln x^\alpha$ for $s \in (0, 1)$ Then, $\varrho$ is generalized exponentially $(s, m)$-convex function in the second sense, for all $\theta \leq -1$, but not a generalized $s$-convex function in the second sense.*

**Remark 1.** *Note that the Definition 10 generalizes and extends some generalized concepts of convexity previously introduced in the literature. In fact, there are special cases such as:*

1. *If $s = 1$, then we have*

$$\varrho(t\iota_1 + m(1 - t)\iota_2) \leq t^\alpha \frac{\varrho(\iota_1)}{\exp^{\theta\iota_1}} + m^\alpha (1 - t)^\alpha \frac{\varrho(\iota_2)}{\exp^{\theta\iota_2}},$$

   *which is called a generalized exponentially $m$-convex function on fractal sets.*
2. *If $m = 1$, then*

$$\varrho(t\iota_1 + (1 - t)\iota_2) \leq t^{\alpha s} \frac{\varrho(\iota_1)}{\exp^{\theta\iota_1}} + (1 - t)^{\alpha s} \frac{\varrho(\iota_2)}{\exp^{\theta\iota_2}},$$

   *which is called a generalized exponentially $s$-convex function on fractal sets.*
3. *If $m = 1$ and $s = 1$, then*

$$\varrho(t\iota_1 + (1 - t)\iota_2) \leq t^\alpha \frac{\varrho(\iota_1)}{\exp^{\theta\iota_1}} + (1 - t)^\alpha \frac{\varrho(\iota_2)}{\exp^{\theta\iota_2}},$$

   *which is called a generalized exponentially convex function on fractal sets.*
4. *If $s = 1$ and $\alpha = 1$, then we have*

$$\varrho(t\iota_1 + m(1 - t)\iota_2) \leq t \frac{\varrho(\iota_1)}{\exp^{\theta\iota_1}} + m(1 - t) \frac{\varrho(\iota_2)}{\exp^{\theta\iota_2}},$$

   *which is called an exponentially $m$-convex function.*

5.  *If $m = 1$ and $\alpha = 1$, then*

$$\varrho(t\iota_1 + (1-t)\iota_2) \leq t^s \frac{\varrho(\iota_1)}{\exp^{\theta\iota_1}} + (1-t)^s \frac{\varrho(\iota_2)}{\exp^{\theta\iota_2}},$$

*which is known as an exponentially s-convex function; see [6].*

6.  *If $m = s = 1$ and $\alpha = 1$, then*

$$\varrho(t\iota_1 + (1-t)\iota_2) \leq t \frac{\varrho(\iota_1)}{\exp^{\theta\iota_1}} + (1-t) \frac{\varrho(\iota_2)}{\exp^{\theta\iota_2}},$$

*which is said to be an exponentially convex function; see [5].*

7.  *Now, if $\theta = 0$, then*

$$\varrho(t\iota_1 + m(1-t)\iota_2) \leq t^{\alpha s}\varrho(\iota_1) + m^\alpha(1-t)^{\alpha s}\varrho(\iota_2),$$

*which is the generalized m-convex functions on fractal sets; see [14].*

8.  *Similarly, if $\alpha = 1$, then we have*

$$\varrho(t\iota_1 + m(1-t)\iota_2) \leq t^s \frac{\varrho(\iota_1)}{\exp^{\theta\iota_1}} + m(1-t)^s \frac{\varrho(\iota_2)}{\exp^{\theta\iota_2}},$$

*which is an exponential (s, m)-convex function in the second sense; see [8].*

9.  *If $m = 1$ and $\theta = 0$, then we obtain*

$$\varrho(t\iota_1 + (1-t)\iota_2) \leq t^{\alpha s}\varrho(\iota_1) + (1-t)^{\alpha s}\varrho(\iota_2),$$

*which is called a generalized s-convex ($0 < s < 1$) in the second sense; see [17].*

10. *If $s = 1$ and $\theta = 0$, then*

$$\varrho(t\iota_1 + (1-t)\iota_2) \leq t^\alpha\varrho(\iota_1) + (1-t)^\alpha\varrho(\iota_2),$$

*which is said to be generalized convex function; see [19].*

11. *If $s = \alpha = 1$ and $\theta = 0$, then*

$$\varrho(t\iota_1 + (1-t)\iota_2) \leq t\varrho(\iota_1) + m(1-t)\varrho(\iota_2),$$

*which is called a m-convex function; see [4].*

12. *If $\alpha = 1$ and $\theta = 0$, then we have*

$$\varrho(t\iota_1 + m(1-t)\iota_2) \leq t^s\varrho(\iota_1) + m(1-t)^s\varrho(\iota_2),$$

*which is an (s, m)-convex function; see [7].*

**Proposition 1.** *Assume that $s, m \in (0, 1]$ and, further let $\varrho_1, \varrho_2 : \mu \longrightarrow \mathbb{R}^\alpha$ be a class of $GE_{s,m}^2$, then*

1.  *$\varrho_1 + \varrho_2$ is a $GE_{s,m}^2$;*
2.  *$\eta^\alpha \varrho_1$ is a $GE_{s,m}^2$.*

**Proof.** Since $\varrho_1$ and $\varrho_2$ are $GE_{s,m}^2$ on $\mu$ and $t \in [0, 1]$, we can obtain

1.

$$(\varrho_1 + \varrho_2)(t\iota_1 + m(1-t)\iota_2) = \varrho_1(t\iota_1 + m(1-t)\iota_2) + \varrho_2(t\iota_1 + m(1-t)\iota_2)$$
$$\leq t^{\alpha s}\frac{\varrho_1(\iota_1)}{\exp^{\theta\iota_1}} + m^\alpha(1-t)^{\alpha s}\frac{\varrho_1(\iota_2)}{\exp^{\theta\iota_2}} + t^{\alpha s}\frac{\varrho_2(\iota_1)}{\exp^{\theta\iota_1}} + m^\alpha(1-t)^{\alpha s}\frac{\varrho_2(\iota_2)}{\exp^{\theta\iota_2}}$$
$$= t^{\alpha s}\frac{\varrho_1(\iota_1) + \varrho_2(\iota_1)}{\exp^{\theta\iota_1}} + m^\alpha(1-t)^{\alpha s}\frac{\varrho_1(\iota_2) + \varrho_2(\iota_2)}{\exp^{\theta\iota_2}}$$
$$= t^{\alpha s}\frac{(\varrho_1 + \varrho_2)(\iota_1)}{\exp^{\theta\iota_1}} + m^\alpha(1-t)^{\alpha s}\frac{(\varrho_1 + \varrho_2)(\iota_2)}{\exp^{\theta\iota_2}}.$$

then $\varrho_1 + \varrho_2$ is a $GE^2_{s,m}$ class on $\mu$.

2.

$$\eta^\alpha \varrho_1(t\iota_1 + m(1-t)\iota_2) \leq \eta^\alpha\left[t^{\alpha s}\frac{\varrho_1(\iota_1)}{\exp^{\theta\iota_1}} + m^\alpha(1-t)^{\alpha s}\frac{\varrho_1(\iota_2)}{\exp^{\theta\iota_2}}\right]$$
$$= t^{\alpha s}\frac{(\eta^\alpha\varrho_1)(\iota_1)}{\exp^{\theta\iota_1}} + m^\alpha(1-t)^{\alpha s}\frac{(\eta^\alpha\varrho_1)(\iota_2)}{\exp^{\theta\iota_2}},$$

hence $\eta^\alpha \varrho_1$ is a class of $GE^2_{s,m}$ on $\mu$. □

**Proposition 2.** *Assume that $\varrho_n : \mu \longrightarrow \mathbb{R}^\alpha$, is a sequence of $GE^2_{s,m}$ pointwise to a function $\varrho : \mu \longrightarrow \mathbb{R}^\alpha$. Then, $\varrho$ is a $GE^2_{s,m}$ on $\mu$.*

**Proof.** Let $\iota_1, \iota_2 \in \mu, t \in [0,1]$ and let $\lim\limits_{n \to \infty} \varrho_n(\iota_1) = \varrho(\iota_1)$, then

$$\varrho(t\iota_1 + m(1-t)\iota_2) = \lim_{n \to \infty}\varrho_n(t\iota_1 + m(1-t)\iota_2)$$
$$\leq \lim_{n \to \infty}\left[t^{\alpha s}\frac{\varrho_n(\iota_1)}{\exp^{\theta\iota_1}} + m^\alpha(1-t)^{\alpha s}\frac{\varrho_n(\iota_2)}{\exp^{\theta\iota_2}}\right]$$
$$= t^{\alpha s}\frac{\lim_{n \to \infty}\varrho_n(\iota_1)}{\exp^{\theta\iota_1}} + m^\alpha(1-t)^{\alpha s}\frac{\lim_{n \to \infty}\varrho_n(\iota_2)}{\exp^{\theta\iota_2}}$$
$$= t^{\alpha s}\frac{\varrho(\iota_1)}{\exp^{\theta\iota_1}} + m^\alpha(1-t)^{\alpha s}\frac{\varrho(\iota_2)}{\exp^{\theta\iota_2}},$$

which means that $\varrho$ is a $GE^2_{s,m}$ on $\mu$. □

**Proposition 3.** *Assume that $s, m \in (0,1]$, if $\varrho_1 : \mu_1 \longrightarrow \mu_2$ is a m-convex and $\varrho_2 : \mu_2 \longrightarrow \mathbb{R}^\alpha$ is a non-decreasing $GE^2_{s,m}$, then $\varrho_2 \circ \varrho_1 : \mu_1 \longrightarrow \mathbb{R}^\alpha$ is a $GE^2_{s,m}$.*

**Proof.** For $\iota_1, \iota_2 \in \mu_1$ and $t \in [0,1]$, we obtain

$$\begin{aligned}(\varrho_2 \circ \varrho_1)(t\iota_1 + m(1-t)\iota_2) &= \varrho_2(\varrho_1(t\iota_1 + m(1-t)y)) \\ &\leq \varrho_2(t\varrho_1(\iota_1) + m(1-t)\varrho_1(\iota_2)) \\ &\leq t^{\alpha s}\frac{\varrho_2(\varrho_1(\iota_1))}{\exp^{\theta\varrho_1(\iota_1)}} + m^\alpha(1-t)^{\alpha s}\frac{\varrho_2(\varrho_1(\iota_2))}{\exp^{\theta\varrho_1(\iota_2)}} \\ &= t^{\alpha s}\frac{(\varrho_2 \circ \varrho_1)(\iota_1)}{\exp^{\theta\varrho_1(\iota_1)}} + m^\alpha(1-t)^{\alpha s}\frac{(\varrho_2 \circ \varrho_1)(\iota_2)}{\exp^{\theta\varrho_1(\iota_2)}}.\end{aligned}$$

Hence, $\varrho_2 \circ \varrho_1$ is a $GE^2_{s,m}$. □

**Theorem 3.** *Assume that $\varrho_1, \varrho_2 : \mu \longrightarrow \mathbb{R}^\alpha$ are both non-negative and monotone increasing. If $\varrho_1, \varrho_2$ are $GE^2_{s,m}$, then $\varrho_1\varrho_2$ is also a $GE^2_{s,m}$.*

**Proof.** If $\iota_1 \leq \iota_2$ (the case $\iota_2 \leq \iota_1$ is similar), then

$$[\varrho_1(\iota_1) - \varrho_1(\iota_2)] \cdot [\varrho_2(\iota_2) - \varrho_2(\iota_1)] \leq 0^\alpha,$$

which implies

$$\varrho_1(\iota_1)\varrho_2(\iota_2) + \varrho_1(\iota_2)\varrho_2(\iota_1) \leq \varrho_1(\iota_1)\varrho_2(\iota_1) + \varrho_1(\iota_2)\varrho_2(\iota_2). \tag{4}$$

On the other hand, for $\iota_1, \iota_2 \in \mu$ and $t \in [0,1]$,

$$
\begin{aligned}
&(\varrho_1\varrho_2)(t\iota_1 + m(1-t)\iota_2)\\
&= \varrho_1(t\iota_1 + m(1-t)\iota_2)\varrho_2(t\iota_1 + m(1-t)\iota_2)\\
&\leq \left[ t^{\alpha s}\frac{\varrho_1(\iota_1)}{\exp^{\theta\iota_1}} + m^\alpha(1-t)^{\alpha s}\frac{\varrho_1(\iota_2)}{\exp^{\theta\iota_2}} \right]\left[ t^{\alpha s}\frac{\varrho_2(\iota_1)}{\exp^{\theta\iota_1}} + m^\alpha(1-t)^{\alpha s}\frac{\varrho_2(\iota_2)}{\exp^{\theta\iota_2}} \right]\\
&= t^{\alpha s}\frac{\varrho_1(\iota_1)\varrho_2(\iota_1)}{\exp^{2\theta\iota_1}} + m^\alpha t^{\alpha s}(1-t)^{\alpha s}\frac{\varrho_1(\iota_1)\varrho_2(\iota_2)}{\exp^{\theta\iota_1}\exp^{\theta\iota_2}}\\
&\quad + m^\alpha t^{\alpha s}(1-t)^{\alpha s}\frac{\varrho_1(\iota_2)\varrho_2(\iota_1)}{\exp^{\theta\iota_1}\exp^{\theta\iota_2}} + m^\alpha(1-t)^{\alpha s}\frac{\varrho_1(\iota_2)\varrho_2(\iota_2)}{\exp^{2\theta\iota_2}}\\
&= t^{2\alpha s}\frac{\varrho_1(\iota_1)\varrho_2(\iota_1)}{\exp^{2\theta\iota_1}} + m^\alpha t^{\alpha s}(1-t)^{\alpha s}\left( \frac{\varrho_1(\iota_1)\varrho_2(\iota_2) + \varrho_1(\iota_2)\varrho_2(\iota_1)}{\exp^{\theta\iota_1}\exp^{\theta\iota_2}} \right)\\
&\quad + m^{2\alpha}(1-t)^{2\alpha s}\frac{\varrho_1(\iota_2)\varrho_2(\iota_2)}{\exp^{2\theta\iota_2}}.
\end{aligned}
$$

On using (4), we obtain

$$
\begin{aligned}
&(\varrho_1\varrho_2)(t\iota_1 + m(1-t)\iota_2)\\
&\leq t^{2\alpha s}\frac{\varrho_1(\iota_1)\varrho_2(\iota_1)}{\exp^{2\theta\iota_1}} + m^\alpha t^{\alpha s}(1-t)^{\alpha s}\left( \frac{\varrho_1(\iota_1)\varrho_2(\iota_1) + \varrho_1(\iota_2)\varrho_2(\iota_2)}{\exp^{\theta\iota_1}\exp^{\theta\iota_2}} \right)\\
&\quad + m^{2\alpha}(1-t)^{2\alpha s}\frac{\varrho_1(\iota_2)\varrho_2(\iota_2)}{\exp^{2\theta\iota_2}}\\
&= \frac{t^{\alpha s}}{\exp^{\theta\iota_1}}\left[ \frac{t^{\alpha s}}{\exp^{\theta\iota_1}} + \frac{m^\alpha(1-t)^{\alpha s}}{\exp^{\theta\iota_2}} \right]\varrho_1(\iota_1)\varrho_2(\iota_1)\\
&\quad + \frac{m^\alpha(1-t)^{\alpha s}}{\exp^{\theta\iota_2}}\left[ \frac{t^{\alpha s}}{\exp^{\theta\iota_1}} + \frac{m^\alpha(1-t)^{\alpha s}}{\exp^{\theta\iota_2}} \right]\varrho_1(\iota_2)\varrho_2(\iota_2).
\end{aligned}
$$

Since $\dfrac{t^{\alpha s}}{\exp^{\theta\iota_1}} + \dfrac{m^\alpha(1-t)^{\alpha s}}{\exp^{\theta\iota_2}} \leq 1^\alpha$, then

$$
\begin{aligned}
(\varrho_1\varrho_2)(t\iota_1 + m(1-t)\iota_2) &\leq t^{\alpha s}\frac{\varrho_1(\iota_1)\varrho_2(x)}{\exp^{\theta\iota_1}} + m^\alpha(1-t)^{\alpha s}\frac{\varrho_1(\iota_2)\varrho_2(\iota_2)}{\exp^{\theta\iota_2}}\\
&= t^{\alpha s}\frac{(\varrho_1\varrho_2)(\iota_1)}{\exp^{\theta\iota_1}} + m^\alpha(1-t)^{\alpha s}\frac{(\varrho_1\varrho_2)(\iota_2)}{\exp^{\theta\iota_2}}.
\end{aligned}
$$

Hence, $\varrho_1\varrho_2$ is a $GE^2_{s,m}$.　□

**Theorem 4.** *Suppose that* $\varrho : [0,\infty) \longrightarrow \mathbb{R}^\alpha$ *is a* $GE^2_{s,m}$, *such that* $0 \leq \iota_1 < m\iota_2 < \infty$. *If* $\varrho \in \mathbb{C}_\alpha[\iota_1, m\iota_2]$, *then*

$$\frac{\iota_1 I^\alpha_{m\iota_2}\varrho(c)}{(m\iota_2 - \iota_1)^\alpha} + \frac{m\iota_1 I^\alpha_{\iota_2}\varrho(c)}{(\iota_2 - m\iota_1)^\alpha} \leq \frac{(1 + m^\alpha)}{2^\alpha \Gamma(\alpha + 1)}\left[ \frac{\varrho(\iota_1)}{\exp^{\theta\iota_1}} + \frac{\varrho(\iota_2)}{\exp^{\theta\iota_2}} \right], \forall m \in [0,1], s \in (0,1].$$

**Proof.** Since $\varrho$ is $GE_{s,m}^2$, for all $t \in [0,1]$ and $\iota_1, \iota_2 \in \mu$, then

$$\varrho(t\iota_1 + m(1-t)\iota_2) \leq t^{\alpha s} \frac{\varrho(\iota_1)}{\exp^{\theta \iota_1}} + m^{\alpha}(1-t)^{\alpha s} \frac{\varrho(\iota_2)}{\exp^{\theta \iota_2}},$$

$$\varrho(t\iota_2 + m(1-t)\iota_1) \leq t^{\alpha s} \frac{\varrho(\iota_2)}{\exp^{\theta \iota_2}} + m^{\alpha}(1-t)^{\alpha s} \frac{\varrho(\iota_1)}{\exp^{\theta \iota_1}},$$

$$\varrho((1-t)\iota_1 + mt\iota_2) \leq (1-t)^{\alpha s} \frac{\varrho(\iota_1)}{\exp^{\theta \iota_1}} + m^{\alpha} t^{\alpha s} \frac{\varrho(\iota_2)}{\exp^{\theta \iota_2}},$$

and

$$\varrho((1-t)\iota_2 + m(1-t)\iota_1) \leq (1-t)^{\alpha s} \frac{\varrho(\iota_2)}{\exp^{\theta \iota_2}} + m^{\alpha} t^{\alpha s} \frac{\varrho(\iota_1)}{\exp^{\theta \iota_1}}.$$

By adding these inequalities together, we obtain

$$
\begin{aligned}
&\varrho(t\iota_1 + m(1-t)\iota_2) + \varrho(t\iota_2 + m(1-t)\iota_1) \\
&+ \varrho((1-t)\iota_1 + mt\iota_2) + \varrho((1-t)\iota_2 + m(1-t)\iota_1) \\
&\leq \left[ \frac{\varrho(\iota_1)}{\exp^{\theta \iota_1}} + \frac{\varrho(\iota_2)}{\exp^{\theta \iota_2}} \right](1 + m^{\alpha}).
\end{aligned}
\tag{5}
$$

Now, integrating the inequality (5) with respect to $t$ over $(0,1)$, then

$$
\begin{aligned}
&\frac{1}{\Gamma(\alpha+1)}\left[ \int_0^1 \varrho(t\iota_1 + m(1-t)\iota_2)(dt)^{\alpha} + \int_0^1 \varrho(t\iota_2 + m(1-t)\iota_1)(dt)^{\alpha} \right. \\
&\left. + \int_0^1 \varrho((1-t)\iota_1 + mt\iota_2)(dt)^{\alpha} + \int_0^1 \varrho((1-t)\iota_2 + m(1-t)\iota_1)(dt)^{\alpha} \right] \\
&\leq \frac{(1+m^{\alpha})}{\Gamma(\alpha+1)}\left[ \frac{\varrho(\iota_1)}{\exp^{\theta \iota_1}} + \frac{\varrho(\iota_2)}{\exp^{\theta \iota_2}} \right].
\end{aligned}
$$

Hence,

$$\frac{{}_{\iota_1}I_{m\iota_2}^{\alpha}\varrho(c)}{(m\iota_2 - \iota_1)^{\alpha}} + \frac{{}_{m\iota_1}I_{\iota_2}^{\alpha}\varrho(c)}{(\iota_2 - m\iota_1)^{\alpha}} \leq \frac{(1+m^{\alpha})}{2^{\alpha}\Gamma(\alpha+1)}\left[ \frac{\varrho(\iota_1)}{\exp^{\theta \iota_1}} + \frac{\varrho(\iota_2)}{\exp^{\theta \iota_2}} \right].$$

$\square$

*Hermite–Hadamard Type Inequality via Generalized Exponentially $(s,m)$-Convex Functions*

This section describes the generalized HH-inequality of $GE_{s,m}^2$ for local fractional integrals.

**Theorem 5.** *Supposing that $\varrho : \mu \longrightarrow \mathbb{R}^{\alpha}$ is a $GE_{s,m}^2$. If $\varrho^{\alpha} \in \mathbb{C}_{\alpha}[\iota_1, m\iota_2]$ for some $0 \leq \iota_1 < \iota_2$, then*

$$
\begin{aligned}
&\frac{2^{\alpha s}}{\Gamma(\alpha+1)}\varrho\left( \frac{c_1 + mc_2}{2} \right) \\
&\leq \frac{2^{\alpha}}{(mc_2 - c_1)^{\alpha}}\left[ {}_{\frac{c_1+mc_2}{2}}I_{mc_2}^{\alpha}\exp^{-\theta \iota_1}\varrho(\iota_1) + {}_{\frac{c_1}{m}}I_{\frac{c_1+mc_2}{2m}}^{\alpha}\exp^{-\theta \iota_2}\varrho(\iota_2) \right] \\
&\leq \frac{2^{\alpha}}{(c_1 - mc_2)^{\alpha(s+1)}}\left\{ \frac{\varrho(c_1)}{\exp^{\theta c_1}} {}_{mc_2}I_{\frac{c_1+mc_2}{2}}^{\alpha}\frac{(\iota_1 - mc_2)^{\alpha s}}{\exp^{-\theta \iota_1}} + m^{\alpha}\frac{\varrho(c_2)}{\exp^{\theta c_2}} {}_{mc_2}I_{\frac{c_1+mc_2}{2}}^{\alpha}\frac{(c_1 - 2\iota_1)^{\alpha s}}{\exp^{-\theta \iota_1}} \right. \\
&\left. + m^{2\alpha}\left[ \frac{\varrho(c_2)}{\exp^{\theta c_2}} {}_{\frac{c_1+mc_2}{2m}}I_{\frac{c_1}{m}}^{\alpha}\frac{(m\iota_2 - c_1)^{\alpha s}}{\exp^{-\theta \iota_2}} + m^{\alpha(s+1)}\frac{\varrho(\frac{c_1}{m^2})}{\exp^{\theta(\frac{c_1}{m^2})}} {}_{\frac{c_1+mc_2}{2m}}I_{\frac{c_1}{m}}^{\alpha}\frac{(c_2 - \iota_2)^{\alpha s}}{\exp^{-\theta \iota_2}} \right] \right\},
\end{aligned}
$$

$\forall s, m \in (0, 1].$

**Proof.** Now, on using the inequality,

$$\varrho\left(\frac{\iota_1 + m\iota_2}{2}\right) \le \frac{1}{2^{\alpha s}}\left[\frac{\varrho(\iota_1)}{\exp^{\theta\iota_1}} + m^\alpha\frac{\varrho(\iota_2)}{\exp^{\theta\iota_2}}\right].$$

Substituting

$$\iota_1 = \frac{\eta}{2}c_1 + m\frac{2-\eta}{2}c_2,$$

$$\iota_2 = \frac{2-\eta}{2m}c_1 + \frac{\eta}{2}c_2;$$

then, we obtain

$$\frac{2^{\alpha s}}{\Gamma(\alpha+1)}\varrho\left(\frac{c_1 + mc_2}{2}\right) \le \frac{\varrho(\frac{\eta}{2}c_1 + m\frac{2-\eta}{2}c_2)}{\exp^{\theta(\frac{\eta}{2}c_1 + m\frac{2-\eta}{2}c_2)}} + m^\alpha\frac{\varrho(\frac{2-\eta}{2m}c_1 + \frac{\eta}{2}c_2)}{\exp^{\theta(\frac{2-\eta}{2m}c_1 + \frac{\eta}{2}c_2)}}.$$

Now, integration in this respect to $\eta$ over $(0,1)$ yields

$$\frac{2^{\alpha s}}{\Gamma(\alpha+1)}\int_0^1 \varrho\left(\frac{c_1 + mc_2}{2}\right)(d\eta)^\alpha$$

$$\le \frac{1}{\Gamma(\alpha+1)}\int_0^1 \frac{\varrho(\frac{\eta}{2}c_1 + m\frac{2-\eta}{2}c_2)}{\exp^{\theta(\frac{\eta}{2}c_1 + m\frac{2-\eta}{2}c_2)}}(d\eta)^\alpha$$

$$+ \frac{m^\alpha}{\Gamma(\alpha+1)}\int_0^1 \frac{\varrho(\frac{2-\eta}{2m}c_1 + \frac{\eta}{2}c_2)}{\exp^{\theta(\frac{2-\eta}{2m}c_1 + \frac{\eta}{2}c_2)}}(d\eta)^\alpha$$

$$= \frac{1}{\Gamma(\alpha+1)}\int_{\frac{c_1+mc_2}{2}}^{mc_2} \frac{2^\alpha\varrho(\iota_1)}{\exp^{\theta\iota_1}(mc_2 - c_1)^\alpha}(d\iota_1)^\alpha$$

$$+ \frac{1}{\Gamma(\alpha+1)}\int_{\frac{c_1+mc_2}{2}}^{mc_2} \frac{2^\alpha\varrho(\iota_1)}{\exp^{\theta\iota_1}(mc_2 - c_1)^\alpha}(d\iota_1)^\alpha$$

$$= \frac{2^\alpha}{(mc_2 + c_1)^\alpha}\left[{}_{\frac{c_1+mc_2}{2}}I^\alpha_{mc_2}\exp^{-\theta\iota_1}\varrho(\iota_1) + {}_{\frac{c_1}{m}}I^\alpha_{\frac{c_1+mc_2}{2m}}\exp^{-\theta\iota_2}\varrho(\iota_2)\right].$$

Since $\varrho$ is a $GE^2_{s,m}$ for $t \in [0,1]$, then

$$\frac{\varrho(\frac{\eta}{2}c_1 + m\frac{2-\eta}{2}c_2)}{\exp^{\theta(\frac{\eta}{2}c_1 + m\frac{2-\eta}{2}c_2)}} + m^\alpha\frac{\varrho(\frac{2-\eta}{2m}c_1 + \frac{\eta}{2}c_2)}{\exp^{\theta(\frac{2-\eta}{2m}c_1 + \frac{\eta}{2}c_2)}}$$

$$\le \frac{1}{\exp^{\theta(\frac{\eta}{2}c_1) + m\frac{2-\eta}{2}c_2}}\left[(\frac{\eta}{2})^{\alpha s}\frac{\varrho(c_1)}{\exp^{\theta c_1}} + m^\alpha(1 - \frac{\eta}{2})^{\alpha s}\frac{\varrho(c_2)}{\exp^{\theta c_2}}\right]$$

$$+ \frac{m^\alpha}{\exp^{\theta(\frac{2-\eta}{2m}c_1) + \frac{\eta}{2}c_2}}\left[(\frac{\eta}{2})^{\alpha s}\frac{\varrho(c_2)}{\exp^{\theta c_2}} + m^\alpha(1 - \frac{\eta}{2})^{\alpha s}\frac{\varrho(\frac{c_1}{m^2})}{\exp^{\theta(\frac{c_1}{m^2})}}\right],$$

similar to the previous one, integrating this respect to $\eta$ over $(0,1)$, we can obtain the second inequality in (6). □

**Lemma 1** ([23]). *For $m \in (0,1]$, let $\varrho : \mu^\circ \longrightarrow \mathbb{R}^\alpha (\mu^\circ$ is the interior of $\mu$) be a function where $k \in D_\alpha(\mu^\circ)$ and $\varrho^{(\alpha)} \in \mathbb{C}_\alpha[\iota_1, m\iota_2]$ for $\iota_1, \iota_2 \in \mu^\circ$ with $\iota_2 > \iota_1$. Then,*

$$\frac{\Gamma(\alpha+1)}{(m\iota_2 - \iota_1)^\alpha}\left[ {}_{\frac{\iota_1 + m\iota_2}{2}}I^\alpha_{m\iota_2}\varrho(u) + m^{2\alpha}_{\frac{\iota_1 + m\iota_2}{2m}}I^\alpha_{\frac{\iota_1}{m}}\varrho(u)\right]$$

$$-(\frac{1}{2})^\alpha\left[\varrho\left(\frac{\iota_1 + m\iota_2}{2}\right) + m^\alpha\varrho\left(\frac{\iota_1 + m\iota_2}{2m}\right)\right]$$

$$= \frac{(m\iota_2 - \iota_1)^\alpha}{4^\alpha}\left[\frac{1}{\Gamma(\alpha+1)}\int_0^1 \eta^\alpha\varrho^{(\alpha)}\left(\frac{\eta}{2}\iota_1 + m\frac{2-\eta}{2}\iota_2\right)(d\eta)^\alpha\right.$$

$$\left.- \frac{1}{\Gamma(\alpha+1)}\int_0^1 \eta^\alpha\varrho^{(\alpha)}\left(\frac{2-\eta}{2m}\iota_1 + m\frac{\eta}{2}\iota_2\right)(d\eta)^\alpha\right].$$

**Theorem 6.** *For some $s, m \in (0,1]$ and $p, q > 1$ with $\frac{1}{p} + \frac{1}{q} = 1$, suppose that $\varrho : \mu^\circ \longrightarrow \mathbb{R}^\alpha$ is a differentiable function $\mu^\circ$ such that $\varrho^{(\alpha)} \in \mathbb{C}_\alpha[\iota_1, m\iota_2]$. If $|\varrho^{(\alpha)}|^q$ is a $GE^2_{s,m}$ on $\mu$ for $q > 1$, then*

$$\frac{\Gamma(\alpha+1)}{(m\iota_2 - \iota_1)^\alpha}\left[ {}_{\frac{\iota_1 + m\iota_2}{2}}I^\alpha_{m\iota_2}\varrho(u) + m^{2\alpha}_{\frac{\iota_1 + m\iota_2}{2m}}I^\alpha_{\frac{\iota_1}{m}}\varrho(u)\right]$$

$$-(\frac{1}{2})^\alpha\left[\varrho\left(\frac{\iota_1 + m\iota_2}{2}\right) + m^\alpha\varrho\left(\frac{\iota_1 + m\iota_2}{2m}\right)\right]$$

$$\leq \frac{(m\iota_2 - \iota_1)^\alpha}{4^\alpha}\left[\frac{\Gamma(\alpha+1)}{\Gamma(2\alpha+1)}\right]^{\frac{1}{p}}\left\{\left[\left(\frac{1}{2}\right)^{\alpha s}\frac{\Gamma((s+1)\alpha+1)}{\Gamma((s+2)\alpha+1)}\frac{|\varrho^{(\alpha)}(\iota_1)|^q}{\exp^{\theta\iota_1}}\right.\right.$$

$$+ m^\alpha\left(\frac{\Gamma(\alpha+1)}{\Gamma(2\alpha+1)} - (\frac{1}{2})^{\alpha s}\frac{\Gamma((s+1)\alpha+1)}{\Gamma((s+2)\alpha)+1}\right)\frac{|\varrho^{(\alpha)}(\iota_2)|^q}{\exp^{\theta\iota_2}}\right]^{\frac{1}{q}} \qquad (6)$$

$$+ \left[m^\alpha\left(\frac{\Gamma(\alpha+1)}{\Gamma(2\alpha+1)} - (\frac{1}{2})^{\alpha s}\frac{\Gamma((s+1)\alpha)+1}{\Gamma((s+2)\alpha)+1}\right)\frac{|\varrho^{(\alpha)}(\frac{\iota_1}{m^2})|^q}{\exp^{\theta(\frac{\iota_1}{m^2})}}\right.$$

$$\left.\left.+ (\frac{1}{2})^{\alpha s}\frac{\Gamma((s+1)\alpha)+1}{\Gamma((s+2)\alpha)+1}\frac{|\varrho^{(\alpha)}(\iota_2)|^q}{\exp^{\theta\iota_2}}\right]^{\frac{1}{q}}\right\}.$$

**Proof.** From Lemma 1, we obtain the next inequality

$$\frac{\Gamma(\alpha+1)}{(m\iota_2 - \iota_1)^\alpha}\left[ {}_{\frac{\iota_1 + m\iota_2}{2}}I^\alpha_{m\iota_2}\varrho(u) + m^{2\alpha}_{\frac{\iota_1 + m\iota_2}{2m}}I^\alpha_{\frac{\iota_1}{m}}\varrho(u)\right]$$

$$-(\frac{1}{2})^\alpha\left[\varrho\left(\frac{\iota_1 + m\iota_2}{2}\right) + m^\alpha\varrho\left(\frac{\iota_1 + m\iota_2}{2m}\right)\right]$$

$$\leq \frac{(m\iota_2 - \iota_1)^\alpha}{4^\alpha}\left[\frac{1}{\Gamma(\alpha+1)}\int_0^1 \eta^\alpha|\varrho^{(\alpha)}\left(\frac{\eta}{2}\iota_1 + m\frac{2-\eta}{2}\iota_2\right)|(d\eta)^\alpha\right.$$

$$\left.+ \frac{1}{\Gamma(\alpha+1)}\int_0^1 \eta^\alpha|\varrho^{(\alpha)}\left(\frac{2-\eta}{2m}\iota_1 + \frac{\eta}{2}\iota_2\right)|(d\eta)^\alpha\right] \qquad (7)$$

$$\leq \frac{(m\iota_2 - \iota_1)^\alpha}{4^\alpha}\left[\frac{\Gamma(\alpha+1)}{\Gamma(2\alpha+1)}\right]^{\frac{1}{p}}$$

$$\times\left\{\left[\frac{1}{\Gamma(\alpha+1)}\int_0^1 \eta^\alpha\left((\frac{\eta}{2})^{\alpha s}\frac{|\varrho^{(\alpha)(\iota_1)}|^q}{\exp^{\theta\iota_1}} + m^\alpha(\frac{2-\eta}{2})^{\alpha s}\frac{|\varrho^{(\alpha)}(\iota_2)|^q}{\exp^{\theta\iota_2}}\right)(d\eta)^\alpha\right]^{\frac{1}{q}}\right.$$

$$\left.+ \left[\frac{1}{\Gamma(\alpha+1)}\int_0^1 \eta^\alpha\left(m^\alpha(\frac{2-\eta}{2})^{\alpha s}\frac{|\varrho^{(\alpha)(\frac{\iota_1}{m^2})}|^q}{\exp^{\theta(\frac{\iota_1}{m^2})}} + (\frac{\eta}{2})^{\alpha s}\frac{|\varrho^{(\alpha)}(\iota_2)|^q}{\exp^{\theta\iota_2}}\right)(d\eta)^\alpha\right]^{\frac{1}{q}}\right\}.$$

Since

$$\frac{1}{\Gamma(\alpha+1)} \int_0^1 \eta^\alpha \left(\frac{\eta}{2}\right)^{\alpha s} (d\eta)^\alpha = \left(\frac{1}{2}\right)^{\alpha s} \frac{\Gamma((s+1)\alpha+1)}{\Gamma((s+2)\alpha+1)}, \tag{8}$$

and

$$\frac{1}{\Gamma(\alpha+1)} \int_0^1 \eta^\alpha \left(\frac{2-\eta}{2}\right)^{\alpha s} (d\eta)^\alpha = \frac{\Gamma(\alpha+1)}{\Gamma(2\alpha+1)} - \left(\frac{1}{2}\right)^{\alpha s} \frac{\Gamma((s+1)\alpha+1)}{\Gamma((s+2)\alpha+1)}, \tag{9}$$

on using (8) and (9) in (7), we obtain the inequality (6). $\quad\square$

**Corollary 1.** *If $s = 1$, then we obtain the following inequality:*

$$\frac{\Gamma(\alpha+1)}{(m\iota_2-\iota_1)^\alpha} \left[ {}_{\frac{\iota_1+m\iota_2}{2}} I^\alpha_{m\iota_2} \varrho(u) + m^{2\alpha} {}_{\frac{\iota_1+m\iota_2}{2m}} I^\alpha_{\frac{\iota_1}{m}} \varrho(u) \right]$$

$$- \left(\frac{1}{2}\right)^\alpha \left[ \varrho\left(\frac{\iota_1+m\iota_2}{2}\right) + m^\alpha \varrho\left(\frac{\iota_1+m\iota_2}{2m}\right) \right]$$

$$\leq \frac{(m\iota_2-\iota_1)^\alpha}{4^\alpha} \left[ \frac{\Gamma(\alpha+1)}{\Gamma(2\alpha+1)} \right]^{\frac{1}{p}} \left\{ \left[ \left(\frac{1}{2}\right)^\alpha \frac{\Gamma(2\alpha+1)}{\Gamma(3\alpha+1)} \frac{|\varrho^{(\alpha)}(\iota_1)|^q}{\exp^{\theta\iota_1}} \right. \right.$$

$$\left. + m^\alpha \left( \frac{\Gamma(\alpha+1)}{\Gamma(2\alpha+1)} - \left(\frac{1}{2}\right)^\alpha \frac{\Gamma(2\alpha+1)}{\Gamma(3\alpha+1)} \right) \frac{|\varrho^{(\alpha)}(\iota_2)|^q}{\exp^{\theta\iota_2}} \right]^{\frac{1}{q}}$$

$$+ \left[ m^\alpha \left( \frac{\Gamma(\alpha+1)}{\Gamma(2\alpha+1)} - \left(\frac{1}{2}\right)^\alpha \frac{\Gamma(2\alpha+1)}{\Gamma(3\alpha+1)} \right) \frac{|\varrho^{(\alpha)}(\frac{\iota_1}{m^2})|^q}{\exp^{\theta(\frac{\iota_1}{m^2})}} \right.$$

$$\left. \left. + \left(\frac{1}{2}\right)^\alpha \frac{\Gamma(2\alpha+1)}{\Gamma(3\alpha+1)} \frac{|\varrho^{(\alpha)}(\iota_2)|^q}{\exp^{\theta\iota_2}} \right]^{\frac{1}{q}} \right\}.$$

**Corollary 2.** *If $m = 1$, then we obtain the following inequality:*

$$\frac{\Gamma(\alpha+1)}{(\iota_2-\iota_1)^\alpha} \left[ {}_{\frac{\iota_1+\iota_2}{2}} I^\alpha_b \varrho(u) + {}_{\frac{\iota_1+\iota_2}{2}} I^\alpha_{\iota_1} \varrho(u) \right] - \varrho\left(\frac{\iota_1+\iota_2}{2}\right)$$

$$\leq \frac{(\iota_2-\iota_1)^\alpha}{4^\alpha} \left[ \frac{\Gamma(\alpha+1)}{\Gamma(2\alpha+1)} \right]^{\frac{1}{p}} \left\{ \left[ \left(\frac{1}{2}\right)^{\alpha s} \frac{\Gamma((s+1)\alpha+1)}{\Gamma((s+2)\alpha+1)} \frac{|\varrho^{(\alpha)}(\iota_1)|^q}{\exp^{\theta\iota_1}} \right. \right.$$

$$\left. + \left( \frac{\Gamma(\alpha+1)}{\Gamma(2\alpha+1)} - \left(\frac{1}{2}\right)^{\alpha s} \frac{\Gamma((s+1)\alpha+1)}{\Gamma((s+2)\alpha+1)} \right) \frac{|\varrho^{(\alpha)}(\iota_2)|^q}{\exp^{\theta\iota_2}} \right]^{\frac{1}{q}}$$

$$+ \left[ \left( \frac{\Gamma(\alpha+1)}{\Gamma(2\alpha+1)} - \left(\frac{1}{2}\right)^{\alpha s} \frac{\Gamma((s+1)\alpha+1)}{\Gamma((s+2)\alpha+1)} \right) \frac{|\varrho^{(\alpha)}(\iota_1)|^q}{\exp^{\theta(\iota_1)}} \right.$$

$$\left. \left. + \left(\frac{1}{2}\right)^{\alpha s} \frac{\Gamma((s+1)\alpha+1)}{\Gamma((s+2)\alpha+1)} \frac{|\varrho^{(\alpha)}(\iota_2)|^q}{\exp^{\theta\iota_2}} \right]^{\frac{1}{q}} \right\}.$$

**Corollary 3.** *If $s = m = 1$, then we obtain the following inequality:*

$$
\frac{\Gamma(\alpha+1)}{(\iota_2-\iota_1)^\alpha} \left[ {}_{\frac{\iota_1+\iota_2}{2}} I^\alpha_{\iota_2} \varrho(u) + {}_{\frac{\iota_1+\iota_2}{2}} I^\alpha_{\iota_1} \varrho(u) \right] - \varrho\left( \frac{\iota_1+\iota_2}{2} \right)
$$

$$
\leq \frac{(\iota_2-\iota_1)^\alpha}{4^\alpha} \left[ \frac{\Gamma(\alpha+1)}{\Gamma(2\alpha+1)} \right]^{\frac{1}{p}} \left\{ \left[ \left( \frac{1}{2} \right)^\alpha \frac{\Gamma(2\alpha+1)}{\Gamma(3\alpha+1)} \frac{|\varrho^{(\alpha)}(\iota_1)|^q}{\exp^{\theta\iota_1}} \right.\right.
$$

$$
+ \left. \left( \frac{\Gamma(\alpha+1)}{\Gamma(2\alpha+1)} - \left( \frac{1}{2} \right)^\alpha \frac{\Gamma(2\alpha+1)}{\Gamma(3\alpha+1)} \right) \frac{|\varrho^{(\alpha)}(\iota_2)|^q}{\exp^{\theta\iota_2}} \right]^{\frac{1}{q}}
$$

$$
+ \left[ \left( \frac{\Gamma(\alpha+1)}{\Gamma(2\alpha+1)} - \left( \frac{1}{2} \right)^\alpha \frac{\Gamma(2\alpha+1)}{\Gamma(3\alpha+1)} \right) \frac{|\varrho^{(\alpha)}(\iota_1)|^q}{\exp^{\theta(\iota_1)}} \right.
$$

$$
+ \left. \left. \left( \frac{1}{2} \right)^\alpha \frac{\Gamma(2\alpha+1)}{\Gamma(3\alpha+1)} \frac{|\varrho^{(\alpha)}(\iota_2)|^q}{\exp^{\theta\iota_2}} \right]^{\frac{1}{q}} \right\}.
$$

**Theorem 7.** *For some $s, m \in (0,1]$ and $p, q > 1$ with $\frac{1}{p} + \frac{1}{q} = 1$, suppose that $\varrho : \mu^\circ \longrightarrow \mathbb{R}^\alpha$ is a differentiable function $\mu^\circ$ such that $\varrho^{(\alpha)} \in \mathbb{C}_\alpha[\iota_1, m\iota_2]$. If $|\varrho^{(\alpha)}|^q$ is a $GE^2_{s,m}$ on $\mu$ for $q > 1$, then*

$$
\frac{\Gamma(\alpha+1)}{(m\iota_2-\iota_1)^\alpha} \left[ {}_{\frac{\iota_1+m\iota_2}{2}} I^\alpha_{m\iota_2} \varrho(u) + m^{2\alpha} {}_{\frac{\iota_1+m\iota_2}{2m}} I^\alpha_{\frac{\iota_1}{m}} \varrho(u) \right]
$$

$$
- \left( \frac{1}{2} \right)^\alpha \left[ \varrho\left( \frac{\iota_1+m\iota_2}{2} \right) + m^\alpha \varrho\left( \frac{\iota_1+m\iota_2}{2m} \right) \right]
$$

$$
\leq \frac{(m\iota_2-\iota_1)^\alpha}{4^\alpha} \left[ \frac{\Gamma(1+p\alpha)}{\Gamma((p+1)\alpha+1)} \right]^{\frac{1}{p}} \left\{ \left[ \frac{\Gamma(s\alpha+1)}{\Gamma((s+1)\alpha+1)} \frac{|\varrho^{(\alpha)}(\iota_1)|^q}{\exp^{\theta\iota_1}} \right.\right.
$$

$$
+ m^\alpha \left. \left( 1 - \left( \frac{1}{2} \right)^{\alpha s} \frac{\Gamma(s\alpha+1)}{\Gamma((s+1)\alpha+1)} \right) \frac{|\varrho^{(\alpha)}(\iota_2)|^q}{\exp^{\theta\iota_2}} \right]^{\frac{1}{q}} \tag{10}
$$

$$
+ \left[ m^\alpha \left( 1 - \left( \frac{1}{2} \right)^{\alpha s} \frac{\Gamma(s\alpha+1)}{\Gamma((s+1)\alpha+1)} \right) \frac{|\varrho^{(\alpha)}(\frac{\iota_1}{m^2})|^q}{\exp^{\theta(\frac{\iota_1}{m^2})}} \right.
$$

$$
+ \left. \left. \left( \frac{1}{2} \right)^{\alpha s} \frac{\Gamma(s\alpha+1)}{\Gamma((s+1)\alpha+1)} \frac{|\varrho^{(\alpha)}(\iota_2)|^q}{\exp^{\theta\iota_2}} \right]^{\frac{1}{q}} \right\}.
$$

**Proof.** From Lemma 1, we obtained the following inequality and the generalized exponentially $(s, m)$-convex function on $\mu$ for $q > 1$, then

$$\frac{\Gamma(\alpha+1)}{(m\iota_2-\iota_1)^\alpha}\left[{}_{\frac{\iota_1+m\iota_2}{2}}I^\alpha_{m\iota_2}\varrho(u)+m^{2\alpha}_{\frac{\iota_1+m\iota_2}{2m}}I^\alpha_{\frac{\iota_1}{m}}\varrho(u)\right]$$

$$-(\frac{1}{2})^\alpha\left[\varrho\left(\frac{\iota_1+m\iota_2}{2}\right)+m^\alpha\varrho\left(\frac{\iota_1+m\iota_2}{2m}\right)\right]$$

$$\leq\frac{(m\iota_2-\iota_1)^\alpha}{4^\alpha}\left[\frac{1}{\Gamma(\alpha+1)}\int_0^1\eta^\alpha|\varrho^{(\alpha)}\left(\frac{\eta}{2}\iota_1+m\frac{2-\eta}{2}\iota_2\right)|(d\eta)^\alpha\right.$$

$$+\frac{1}{\Gamma(\alpha+1)}\int_0^1\eta^\alpha|\varrho^{(\alpha)}\left(\frac{2-\eta}{2m}\iota_1+\frac{\eta}{2}\iota_2\right)|(d\eta)^\alpha\bigg]$$

$$\leq\frac{(m\iota_2-\iota_1)^\alpha}{4^\alpha}\left[\frac{1}{\Gamma(\alpha+1)}\int_0^1\eta^{p\alpha}(d\eta)^\alpha\right]^{\frac{1}{p}}$$

$$\times\left[\left(\frac{1}{\Gamma(\alpha+1)}\int_0^1\eta^\alpha|\varrho^{(\alpha)}\left(\frac{\eta}{2}\iota_1+m\frac{2-\eta}{2}\iota_2\right)|^q(d\eta)^\alpha\right)^{\frac{1}{q}}\right.$$

$$+\left(\frac{1}{\Gamma(\alpha+1)}\int_0^1\eta^\alpha|\varrho^{(\alpha)}\left(\frac{2-\eta}{2m}\iota_1+\frac{\eta}{2}\iota_2\right)|^q(d\eta)^\alpha\right)^{\frac{1}{q}}\bigg]$$

$$\leq\frac{(m\iota_2-\iota_1)^\alpha}{4^\alpha}\left[\frac{\Gamma(p\alpha+1)}{\Gamma((p+1)\alpha+1)}\right]^{\frac{1}{p}}$$

$$\times\left\{\left[\frac{1}{\Gamma(\alpha+1)}\int_0^1\left((\frac{\eta}{2})^{\alpha s}\frac{|\varrho^{(\alpha)(\iota_1)}|^q}{\exp^{\theta\iota_1}}+m^\alpha(\frac{2-\eta}{2})^{\alpha s}\frac{|\varrho^{(\alpha)}(\iota_2)|^q}{\exp^{\theta\iota_2}}\right)(d\eta)^\alpha\right]^{\frac{1}{q}}\right.$$

$$+\left[\frac{1}{\Gamma(\alpha+1)}\int_0^1\left(m^\alpha(\frac{2-\eta}{2})^{\alpha s}\frac{|\varrho^{(\alpha)(\frac{\iota_1}{m^2})}|^q}{\exp^{\theta(\frac{\iota_1}{m^2})}}+(\frac{\eta}{2})^{\alpha s}\frac{|\varrho^{(\alpha)}(\iota_2)|^q}{\exp^{\theta\iota_2}}\right)(d\eta)^\alpha\right]^{\frac{1}{q}}\bigg\}$$

$$=\frac{(m\iota_2-\iota_1)^\alpha}{4^\alpha}\left[\frac{\Gamma(p\alpha+1)}{\Gamma((p+1)\alpha+1)}\right]^{\frac{1}{p}}\left\{\left[\frac{\Gamma(s\alpha+1)}{\Gamma((s+1)\alpha+1)}\frac{|\varrho^{(\alpha)}(\iota_1)|^q}{\exp^{\theta\iota_1}}\right.\right.$$

$$+m^\alpha\left(1-(\frac{1}{2})^{\alpha s}\frac{\Gamma(s\alpha+1)}{\Gamma((s+1)\alpha+1)}\right)\frac{|\varrho^{(\alpha)}(\iota_2)|^q}{\exp^{\theta\iota_2}}\bigg]^{\frac{1}{q}}$$

$$+\left[m^\alpha\left(1-(\frac{1}{2})^{\alpha s}\frac{\Gamma(s\alpha+1)}{\Gamma((s+1)\alpha+1)}\right)\frac{|\varrho^{(\alpha)}(\frac{\iota_1}{m^2})|^q}{\exp^{\theta(\frac{\iota_1}{m^2})}}\right.$$

$$+(\frac{1}{2})^{\alpha s}\frac{\Gamma(s\alpha+1)}{\Gamma((s+1)\alpha+1)}\frac{|\varrho^{(\alpha)}(\iota_2)|^q}{\exp^{\theta\iota_2}}\bigg]^{\frac{1}{q}}\bigg\}.$$

□

Some particular cases of the last Theorem will be presented

**Corollary 4.** *If $s = 1$, then we obtain the following inequality:*

$$\frac{\Gamma(\alpha+1)}{(m\iota_2-\iota_1)^\alpha}\left[\,_{\frac{\iota_1+m\iota_2}{2}}I^\alpha_{m\iota_2}\varrho(u)+m^{2\alpha}\,_{\frac{\iota_1+m\iota_2}{2m}}I^\alpha_{\frac{\iota_1}{m}}\varrho(u)\right]-(\frac{1}{2})^\alpha\left[\varrho\left(\frac{\iota_1+m\iota_2}{2}\right)+m^\alpha\varrho\left(\frac{\iota_1+m\iota_2}{2m}\right)\right]$$

$$\leq\frac{(m\iota_2-\iota_1)^\alpha}{4^\alpha}\left[\frac{\Gamma(\alpha+1)}{\Gamma(2\alpha+1)}\right]^{\frac{1}{p}}\left\{\left[\frac{\Gamma(\alpha+1)}{\Gamma(2\alpha+1)}\frac{|\varrho^{(\alpha)}(\iota_1)|^q}{\exp^{\theta\iota_1}}\right.\right.$$

$$+m^\alpha\left(1-(\frac{1}{2})^\alpha\frac{\Gamma(\alpha+1)}{\Gamma(2\alpha+1)}\right)\frac{|\varrho^{(\alpha)}(\iota_2)|^q}{\exp^{\theta\iota_2}}\Bigg]^{\frac{1}{q}}$$

$$+\left[m^\alpha\left(1-(\frac{1}{2})^\alpha\frac{\Gamma(2\alpha+1)}{\Gamma(3\alpha+1)}\right)\frac{|\varrho^{(\alpha)}(\frac{\iota_1}{m^2})|^q}{\exp^{\theta(\frac{\iota_1}{m^2})}}\right.$$

$$\left.\left.+(\frac{1}{2})^\alpha\frac{\Gamma(1+\alpha)}{\Gamma(2\alpha+1)}\frac{|\varrho^{(\alpha)}(\iota_2)|^q}{\exp^{\theta\iota_2}}\right]^{\frac{1}{q}}\right\}.$$

**Corollary 5.** *If $m = 1$, then we obtain the following inequality:*

$$|\frac{\Gamma(\alpha+1)}{(\iota_2-\iota_1)^\alpha}\left[\,_{\frac{\iota_1+\iota_2}{2}}I^\alpha_{\iota_2}\varrho(u)+\,_{\frac{\iota_1+\iota_2}{2}}I^\alpha_{\iota_1}\varrho(u)\right]-\varrho\left(\frac{\iota_1+\iota_2}{2}\right)|$$

$$\leq\frac{(\iota_2-\iota_1)^\alpha}{4^\alpha}\left[\frac{\Gamma(p\alpha+1)}{\Gamma((p+1)\alpha+1)}\right]^{\frac{1}{p}}\left\{\left[\frac{\Gamma(s\alpha+1)}{\Gamma((s+1)\alpha+1)}\frac{|\varrho^{(\alpha)}(\iota_1)|^q}{\exp^{\theta\iota_1}}\right.\right.$$

$$+\left(1-(\frac{1}{2})^{\alpha s}\frac{\Gamma(s\alpha+1)}{\Gamma(1+(1+s)\alpha)}\right)\frac{|\varrho^{(\alpha)}(\iota_2)|^q}{\exp^{\theta\iota_2}}\Bigg]^{\frac{1}{q}}$$

$$+\left[\left(1-(\frac{1}{2})^{\alpha s}\frac{\Gamma(s\alpha+1)}{\Gamma((s+1)\alpha+1)}\right)\frac{|\varrho^{(\alpha)}(\iota_1)|^q}{\exp^{\theta\iota_1}}\right.$$

$$\left.\left.+(\frac{1}{2})^{\alpha s}\frac{\Gamma(s\alpha+1)}{\Gamma((s+1)\alpha+1)}\frac{|\varrho^{(\alpha)}(\iota_2)|^q}{\exp^{\theta\iota_2}}\right]^{\frac{1}{q}}\right\}.$$

**Corollary 6.** *If $s = m = 1$, then we obtain the following inequality:*

$$\frac{\Gamma(\alpha+1)}{(\iota_2-\iota_1)^\alpha}\left[\,_{\frac{\iota_1+\iota_2}{2}}I^\alpha_{\iota_2}\varrho(u)+\,_{\frac{\iota_1+\iota_2}{2}}I^\alpha_{\iota_1}\varrho(u)\right]-\varrho\left(\frac{\iota_1+\iota_2}{2}\right)$$

$$\leq\frac{(\iota_2-\iota_1)^\alpha}{4^\alpha}\left[\frac{\Gamma(p\alpha+1)}{\Gamma((p+1)\alpha+1)}\right]^{\frac{1}{p}}\left\{\left[\frac{\Gamma(\alpha+1)}{\Gamma(2\alpha+1)}\frac{|\varrho^{(\alpha)}(\iota_1)|^q}{\exp^{\theta\iota_1}}\right.\right.$$

$$+\left(1-(\frac{1}{2})^\alpha\frac{\Gamma(\alpha+1)}{\Gamma(2\alpha+1)}\right)\frac{|\varrho^{(\alpha)}(\iota_2)|^q}{\exp^{\theta\iota_2}}\Bigg]^{\frac{1}{q}}$$

$$+\left[\left(1-(\frac{1}{2})^{\alpha s}\frac{\Gamma(\alpha+1)}{\Gamma(2\alpha+1)}\right)\frac{|\varrho^{(\alpha)}(\iota_1)|^q}{\exp^{\theta(\iota_1)}}\right.$$

$$\left.\left.+(\frac{1}{2})^\alpha\frac{\Gamma(\alpha+1)}{\Gamma(2\alpha+1)}\frac{|\varrho^{(\alpha)}(\iota_2)|^q}{\exp^{\theta\iota_2}}\right]^{\frac{1}{q}}\right\}.$$

## 3. Applications

### 3.1. Applications to Special Means

Let us consider $\alpha$-type special means; see [24]. For two positive real numbers, $\iota_1, \iota_2$ where $\iota_1 < \iota_2$

1. the generalized arithmetic $A_\alpha(\iota_1, \iota_2) = \left(\frac{\iota_1 + \iota_2}{2}\right)^\alpha = \frac{\iota_1^\alpha + \iota_2^\alpha}{2^\alpha}$.

2. $L_\alpha(\iota_1, \iota_2) = \left[\frac{\Gamma(1+k\alpha)}{\Gamma(1+(k+1)\alpha)}\right], k \in \mathbb{Z}\{-1, 0\}$ and $\iota_1, \iota_2 \in \mathbb{R}$ with $a \neq b$.

   Let $\varrho(\iota_1) = \iota_1^{k\alpha}(\varrho \in \mathbb{R}, k \in \mathbb{Z}, |k| \geq 2)$; then,

1. By applying in Corollary 5, we obtain the next result:

$$
\left| \frac{\Gamma(\alpha+1)}{(\iota_2 - \iota_1)^\alpha} \left[ L_{k\alpha}^k(\iota_2, A(\iota_1, \iota_2)) + L_{k\alpha}^k(A(\iota_1, \iota_2), \iota_1) \right] - A_\alpha^k(\iota_1, \iota_2) \right|
$$

$$
\leq \frac{(\iota_2 - \iota_1)^\alpha}{4^\alpha} \left[ \frac{\Gamma(p\alpha+1)}{\Gamma((p+1)\alpha+1)} \right]^{\frac{1}{p}} \left\{ \left[ \frac{\Gamma(s\alpha+1)}{\Gamma((s+1)\alpha+1)} \left( \frac{\Gamma(1+k\alpha)}{\Gamma(1+(k-1)\alpha)} \right)^q \frac{|\iota_1^{(k-1)\alpha}|^q}{\exp^{\theta\iota_1}} \right. \right.
$$

$$
+ \left. \left(1 - (\tfrac{1}{2})^{\alpha s} \frac{\Gamma(s\alpha+1)}{\Gamma(1+(1+s)\alpha)} \right) \left( \frac{\Gamma(1+k\alpha)}{\Gamma(1+(k-1)\alpha)} \right)^q \frac{|\iota_2^{(k-1)\alpha}|^q}{\exp^{\theta\iota_2}} \right]^{\frac{1}{q}}
$$

$$
+ \left[ \left(1 - (\tfrac{1}{2})^{\alpha s} \frac{\Gamma(s\alpha+1)}{\Gamma((s+1)\alpha+1)} \right) \left( \frac{\Gamma(1+k\alpha)}{\Gamma(1+(k-1)\alpha)} \right)^q \frac{|\iota_1^{(k-1)\alpha}|^q}{\exp^{\theta\iota_1}} \right.
$$

$$
+ \left. \left. (\tfrac{1}{2})^{\alpha s} \frac{\Gamma(s\alpha+1)}{\Gamma((s+1)\alpha+1)} \left( \frac{\Gamma(1+k\alpha)}{\Gamma(1+(k-1)\alpha)} \right)^q \frac{|\iota_2^{(k-1)\alpha}|^q}{\exp^{\theta\iota_2}} \right]^{\frac{1}{q}} \right\}.
$$

2. By applying in Corollary 6, we obtain the next result:

$$
\left| \frac{\Gamma(\alpha+1)}{(\iota_2 - \iota_1)^\alpha} \left[ L_{k\alpha}^k(\iota_2, A(\iota_1, \iota_2)) + L_{k\alpha}^k(A(\iota_1, \iota_2), \iota_1) \right] - A_\alpha^k(\iota_1, \iota_2) \right|
$$

$$
\leq \frac{(\iota_2 - \iota_1)^\alpha}{4^\alpha} \left[ \frac{\Gamma(p\alpha+1)}{\Gamma((p+1)\alpha+1)} \right]^{\frac{1}{p}} \left\{ \left[ \frac{\Gamma(\alpha+1)}{\Gamma(2\alpha+1)} \left( \frac{\Gamma(1+k\alpha)}{\Gamma(1+(k-1)\alpha)} \right)^q \frac{|\iota_1^{(k-1)\alpha}|^q}{\exp^{\theta\iota_1}} \right. \right.
$$

$$
+ \left. \left(1 - (\tfrac{1}{2})^\alpha \frac{\Gamma(\alpha+1)}{\Gamma(2\alpha+1)} \right) \left( \frac{\Gamma(1+k\alpha)}{\Gamma(1+(k-1)\alpha)} \right)^q \frac{|\iota_2^{(k-1)\alpha}|^q}{\exp^{\theta\iota_2}} \right]^{\frac{1}{q}}
$$

$$
+ \left[ \left(1 - (\tfrac{1}{2})^{\alpha s} \frac{\Gamma(\alpha+1)}{\Gamma(2\alpha+1)} \right) \left( \frac{\Gamma(1+k\alpha)}{\Gamma(1+(k-1)\alpha)} \right)^q \frac{|\iota_1^{(k-1)\alpha}|^q}{\exp^{\theta\iota_1}} \right.
$$

$$
+ \left. \left. (\tfrac{1}{2})^\alpha \frac{\Gamma(\alpha+1)}{\Gamma(2\alpha+1)} \left( \frac{\Gamma(1+k\alpha)}{\Gamma(1+(k-1)\alpha)} \right)^q \frac{|\iota_2^{(k-1)\alpha}|^q}{\exp^{\theta\iota_2}} \right]^{\frac{1}{q}} \right\}.
$$

### 3.2. Inequalities for Some Special Functions

For $q \in (0, 1)$, the $q$-digamma function $\omega_q$ is defined by

$$
\omega_q(\iota) = -\ln(1-q) + \ln q \sum_{i=0}^{\infty} \frac{q^{i+\iota}}{1 - q^{i+\iota}}
$$

$$
-\ln(1-q) + \ln \sum_{i=1}^{\infty} \frac{q^{i\iota}}{1 - q^i}.
$$

In addition, for $q > 1$, the $q$-digamma function $\omega_q$ is defined by

$$
\omega_q(\iota) = -\ln(q-1) + \ln q \left[ \iota - \frac{1}{2} - \sum_{i=0}^{\infty} \frac{q^{-(i+\iota)}}{1 - q^{-(i+\iota)}} \right]
$$

$$
-\ln(q-1) + \ln q \left[ \iota - \frac{1}{2} - \sum_{i=1}^{\infty} \frac{q^{-(i\iota)}}{1 - q^{-(i\iota)}} \right].
$$

If we set $\varrho(\iota) = \omega'_q$ in Corollary 3, then we have the following inequality:

$$\frac{1}{(\iota_2 - \iota_1)}\left[\omega(\iota_2) - \omega\left(\frac{\iota_1 + \iota_2}{2}\right)\right] - \omega'\left(\frac{\iota_1 + \iota_2}{2}\right)$$

$$\leq \frac{(\iota_2 - \iota_1)}{4}\left[\frac{1}{2}\right]^{\frac{1}{p}}\left\{\left[\left(\frac{1}{3}\right)\frac{|\omega''(\iota_1)|^q}{\exp^{\theta\iota_1}} + \left(\frac{1}{6}\right)\frac{|\omega''(\iota_2)|^q}{\exp^{\theta\iota_2}}\right]^{\frac{1}{q}}\right.$$

$$\left. + \left[\left(\frac{1}{6}\right)\frac{|\omega''(\iota_1)|^q}{\exp^{\theta(\iota_1)}} + \left(\frac{1}{3}\right)\frac{|\omega''(\iota_2)|^q}{\exp^{\theta\iota_2}}\right]^{\frac{1}{q}}\right\}.$$

## 4. Conclusions

In this work, we discussed the generalized exponential $(s, m)$-convex function, a novel concept for the generalized $(s, m)$-convex function. The proposed definition's algebraic properties were looked at. We described the innovative Hermite–Hadamard type inequality in line with the newly proposed concept. We also developed a few theorems. The new family of $(s, m)$ functions can be thought of as a noteworthy expansion and refinement of our obtained results in the new theorems. In addition, all the results of this paper hold for $s$-convex, $m$-convex, exponentially convex, exponentially $s$-convex, and convex functions by taking special cases. In particular, it results in generalized exponential $(s, m)$-convex functions, which are proved in [8], being able to be obtained. Applications to unique means were taken into account. We also came up with some fascinating and amazing similarities. In addition, in future work, we will study Fejér–Hadamard Inequalities associated with generalized exponential $(s, m)$-convex functions and new inequalities via $n$-polynomial generalized exponential $(s, m)$-convex functions.

**Author Contributions:** Both authors thought of the study, contributed to its conception and coordination, prepared the manuscript, and reviewed and approved the final version. All authors have read and agreed to the published version of the manuscript.

**Funding:** This research received no external funding.

**Data Availability Statement:** This study did not report any data.

**Acknowledgments:** The authors thank the referees and the editor for very constructive and valuable suggestions that improved the current manuscript.

**Conflicts of Interest:** The authors declare no conflict of interest.

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
