# Peer review of "Some Local Fractional Inequalities Involving Fractal Sets via Generalized Exponential (s,m)-Convexity"

_axioms, doi:10.3390/axioms12020106_

Round 1
Author Response
First Reviewer Reports:
In this paper, the authors introduced the concept of generalized exponentially (s, m)-convex functions, and to determine some properties. They also studied the interactions between generalized exponentially (s, m)-convex functions and local fractional integrals. .After checking the full article carefully, I recommended this paper to publish in Axioms after the following
- On page 3, line -4. The authors should explain the concept of “ local fractional continuous”
We have rewritten it.
2- On page 3, lines -4 and-3. “The local fractional function” should be replaced by “The local fractional integral of function”.
We have replaced it.
- On page 8, line -5. “ ” should be " ".
We have rewritten it.
4- On page 9, line -3. “ ” should be modified
We have modified it.
5- On page 11, line 9. “ ” " "should be replaced by " "
We have replaced it.
6- On page 12, lines -7 and -1. “ ” should be modified.
We have modified it.
7- On page 16, line 9. “ n” should be deleted.
We have deleted it.
- The authors had better give some remarks to illustrate the significance of their conclusions
We have rewritten a conclusion. Also, some explanations have been added.
Dear Professor,
First of all, the authors appreciate all your remarks and we would like to express our sincere thanks to the referee for the careful and detailed reading of the manuscript and suggestions on the current literature. We have applied all your comments and revised the manuscript accordingly attached the revised version for your further kind consideration,
Sincerely,
Reviewer 2 Report
For the major modifications, find the attached pdf.

Author Response
Second Reviewer Reports:
I accept this paper after major modifications.
- Title The title of your manuscript is usually the first introduction readers have to your published work. Therefore, you must select a title that grabs attention, accurately describes the contents of your manuscript, and makes people want to read further. So “Some new local fractional inequalities on fractal sets and generalized exponentially (s, m)-convex functions” should be re-placed by “Some local fractional inequalities involving fractal sets via generalized exponential (s, m)-convexity”.
We have replaced it.
- Abstract A concise and factual abstract is always required in a scientific paper. The abstract should state briefly the purpose of the research, the principal results, and major conclusions. In this paper, Abstract in the current form is not enough. Must Revise.
We have rewritten it.
- Keywords Mentioned Keywords ( Harmonically convex; Exponentially harmonically s-convex functions; Fractal sets) details are missing. Add details or remove these words. Because Keywords are a tool to help indexers and search engines find relevant papers. If database search engines can find your journal manuscript, readers will be able to find it too. This will increase the number of people reading your manuscript, and likely lead to more citations.
We have rewritten it.
- Introduction
1) The opening paragraph of the paper will provide paper readers with their initial impressions about the logic of the paper argument, writing style, the overall quality of research, and, ultimately, the validity of your findings and conclusions. Here, the introduction is required to improve by acknowledging some previous works. Also, in my opinion, the authors should add some more appropriate work in this direction and highlighting their contributions. so section 1 is not enough. More recent results/development should be presented to support this research. Some applications are also needed in Section 1.
2) Highlight the aim and novelty of your work.
We have done that.
3) Revise references according to the ascending order.
We have rewritten them.
4) Add the “paper construction” paragraph in the last of introduction section.
We have added.
5) To improve the impact and readership of your manuscript, the author needs to clearly articulate in the Abstract and in the Introduction sections about the uniqueness or novelty of this paper, and why or how it is different from other similar papers.
We have rewritten it.
- Preliminaries
- Preliminaries section is missing. Add the section “Preliminaries” after introduction section and mentioned all the published definitions or work which are used throughout the paper.
We have added preliminaries with introduction section
- Add the list of abbriavations in the end of “Preliminaries” section.
We have explained any abbreviation the first time we have used it.
"Generalized exponentially (s, m)–convex functions and associated algebraic properties"
- Replace “Generalized exponentially (s, m)–convex functions” by “Generalized exponentially (s, m)–convex functions and associated algebraic properties”.
We have rewritten it.
- For the interest of readers and quality of work, add some discussion regarding the scope of exponentially convex functions in the start of this section.
We have added some discussion at the start of this section.
- Put a dot at the end of Proposition 2.4.
We have put it.
- Add some discussion about the polynomial exponentially (s, m)–convex functions in the second sense.
We have seen that it is not necessary to add some discussion about the polynomial exponentially (s, m)–convex functions in the second sense because this topic needs to add polynomial generalized exponentially (s, m)–convex functions in the second sense which is not in the current work.
- For the definition worth, add some examples regarding Generalized exponentially (s, m)–convex functions.
We have added an example about Generalized exponentially (s, m)–convex functions.
- Every exponential convex function is exponentially (s, m)–convex functions? Prove and add them as a proposition.
This point is not in the current work, this point has been discussed
in Qiang, Xiaoli, et al. "Generalized fractional integral inequalities for exponentially (s, m) $(s, m) $-convex functions." Journal of Inequalities and Applications 2020.1 (2020): 1-13.
- Every exponential convex function is generalized exponentially (s, m)–convex functions? Prove and add them as a proposition.
We have added a remark about the relation between exponential convex function is generalized exponentially (s, m)–convex functions.
- Every exponential convex function is a bounded. Discuss every generalized exponentially (s, m)–convex function is a bounded?.
If we will add this information, then we will study more such as the level set, epi, quasi functions, and so on, so we will leave it as future work.
- In theorem 2.6, “g” replace by .
We have rewritten it.
- In theorem 2.6, “” replace by 0.
We can not do that because we study in fractional space.
- Put comma at the end of every inequality.
We have put a comma at the end of every inequality if it is necessarily.
- In the proof of Proposition 2.3, what is the meaning of “b” .
We have corrected it.
- In Proposition 2.3, the condition for s is missing.
We have put it.
- Line 4, In the proof of Proposition 2.4, what is the meaning of “b” .
We have corrected it.
- In the proof of Proposition 2.4, put comma at the end of last inequality.
We have put it.
- In the proof of Proposition 2.5, the conditions for s and m are missing.
We have put it.
- Line 2, In the proof of Proposition 2.5, what is the meaning of “y” .
We have corrected it.
- In the proof of Proposition 2.6, the conditions for s, α and m are missing.
We have put it.
- Respected authors introduced newly definition namely “generalized exponentially (s, m)–convex functions” for the mentioned conditions s, m ∈ [0, 1], but here in theorem 2.7, why authors use the condition (0, 1] for s. Give a logical reason.
We have changed all of them to (0, 1].
Hermite-Hadamard type inequality via Generalized exponentially (s, m)–convex functions
Replace “2.1 Some New Results on Generalized Exponentially (s, m)-Convex function” by “Hermite-Hadamard type inequality via Generalized exponentially (s, m)–convex functions”.
We have rewritten it.
Main Results
- Make new section namely “Main Results”.
We have added two sections that "Generalized exponentially (s, m)–convex functions and associated algebraic properties "and " Hermite-Hadamard type inequality via Generalized exponentially (s, m)–convex functions" all of them are the main results.
- Respected authors introduced newly definition namely “generalized exponentially (s, m)–convex functions” for the mentioned conditions s, m ∈ [0, 1], but here authors use the already published lemma which are hold for the condition s, m ∈ (0, 1] , why authors use the condition (0, 1] for s, m in the theorems. Give a logical reason.
We have changed all of them to (0, 1].
- Why authors are doing this work? What is the hurdle encountered by this study?
We have defined and introduced some new concepts of convex functions called generalized exponential ( s,m )-convex functions on fractal space, and explore some of their properties that are generalized to exponentially (s, m)-convex functions, and explore some of their properties. Results obtained in this paper can be viewed as refinement and improvement of previously known.
- The authors should also check for any grammatical errors throughout the paper.
We have corrected a lots of grammar mistakes throughout the paper, and please see the details.
- A lot of typo errors, I faced. Remove them throughout the paper.
We have removed all type errors.
- No clear motivation and reason to study such type of fractional inequalities has proved in pure and applied mathematics. So for quality and interest of readers, add some advantages of the study ?.
Here
- Remove the unnecessary equation numbers throughout the paper.
We have removed all type errors.
- Add some possible applications and future directions.
We have added.
- At page 13, remove space.
We have removed it.
- At page 14, some lines are cross the page limit. Manage them. infinitely many solutions for fractional Laplacian problems with local growth conditions.
We have rewritten them.
- Statement of the Corollarly 2.11 and 2.15 are replace by “ If s = 1, then we obtain the following inequality”.
We have rewritten them.
- Statement of the Corollarly 2.12 and 2.16 are replace by “ If m = 1, then we obtain the following inequality”.
We have rewritten them.
- Statement of the Corollarly 2.13 and 2.17 are replace by “ If s = m = 1, then we obtain the following inequality”.
We have rewritten them.
- For the interest of readers and quality of work, must add some examples in the sence of symmetry using the above theorems.
We have added some applications that are used through some of the special cases in some theorems.
Limitations of the research
- Please include a new section to discuss about the ”Limitations of the research” because it is currently missing.
Here
- Add the scope of newly introduced concept. For example, Improved Holder and Improved power mean inequality gives the best result as compare to Holder and power inequality. Similarly, generalized exponential (s, m)–convex function is the modified version of exponential (s, m)–convex function. Is generalized exponential (s, m)–convex function gives a best result as compare to exponential (s, m)–convex function? Give a logical reasons for the dear readers.
We have rewritten this in the introduction so that readers understand the paper's flow and purpose through the infiltration of concepts.
Applications
- Application section in the current form is not enough. Must Revise.
We have added some applications.
- Add some applications related q-digamma and quadrature type.
We have tried to add some applications related to another type.
Conclusion
- The Conclusion section is too short. Must revise. To inspire the readers, please provide a few more examples of how ” generalized exponential (s, m)–convex function” can be applied in the real world.
We have rewritten it.
- In conclusion, possible applications and future directions should be added.
We have rewritten it.
References
References are ok. But here acceptable in the format of MDPI template
We have rewritten it.
Dear Professor,
First of all, the authors appreciate all the remarks very much, so we would like to express our sincere thanks to the referee for the careful and detailed reading of the manuscript and suggestions on the current literature. We have applied all your comments and revised the manuscript according to your comments and attached for your further kind consideration,
Sincerely

Reviewer 3 Report
Report on the paper
Some new local fractional inequalities on fractal sets and generalized exponentially (s, m)-convex functions
by Wedad Saleh and Adem Kılıcman
The aim of the authors is to 'explore the concept of generalized exponentially (s,m)-convex functions, and to determine some properties'.
The paper has theoretically valuable content, but consistent examples are missing.
I believe that the applications considered are not enough and the introduction of more concrete examples would increase the value of the paper.
Some formulas in a too-large format must be broken, using for example some notations, or another way.
The section Concluding remarks must be enlarged, pointing out more ideas.
I recommend the publication after fulfilling the above.
Author Response
Third Reviewer Reports:
I recommend the publication after fulfilling the following.
- The paper has theoretically valuable content, but consistent examples are missing.
We have rewritten an example.
- I believe that the applications considered are not enough and the introduction of more concrete examples would increase the value of the paper.
We have added some applications.
- Some formulas in a too-large format must be broken, using for example some notations, or another way.
We have rewritten them.
- The section Concluding remarks must be enlarged, pointing out more ideas.
We have rewritten it.
Dear Professor,
First of all, the authors appreciate all the remarks very much, and so we would like to express our sincere thanks to you for the careful and detailed reading of the manuscript and suggestions on the current literature. We have applied all your comments and revised the manuscript according to your comment and attached for your further kind consideration,
Sincerely,

Reviewer 4 Report
See the attached report.

Author Response
Fourth Reviewer Reports:
During the review, I found that the results presented here are looking meaningful and indeed scientifically and mathematically correct. In my opinion, the paper is sound and interesting in the field of fractional integral inequalities.
Therefore, I recommend the article be accepted for publication in Axioms after providing minor revisions.
Comments and suggestions:
- The author should read and check the full article very carefully to correct possible grammar and spelling mistakes.
We have corrected a lots of grammar mistakes throughout the paper, and you can read it.
- Writing equations in a more orderly manner.
We have rewritten them.
Dear Professor,
First of all, the authors appreciate all the remarks very much, and we would like to express our sincere thanks to the referee for the careful and detailed reading of the manuscript and suggestions on the current literature. We have applied all your comments and revised the manuscript accordingly and attached for your further kind consideration,
Sincerely,

Round 2
Reviewer 2 Report
I recommend this paper for minor revision. For the comments, see the attached file.

Author Response
Dear Professor, Thanks for your constructive and enlightening comments. The authors really appreciate all the remarks very much, thus we would like to express our sincere thanks to the referee for the careful and detailed reading of the manuscript and suggestions on the current literature.
Second Reviewer Reports:
I accept this paper after minor modifications.
- Keywords Mentioned Keywords ( Harmonically convex; Exponentially harmonically s-convex functions; Fractal sets) details are missing. Add details or remove these words. Because Keywords are a tool to help indexers and search engines find relevant papers. If database search engines can find your journal manuscript, readers will be able to find it too. This will increase the number of people reading your manuscript, and likely lead to more citations.
We have rewritten it.
- Introduction
1) Revise references according to the ascending order.
We have done that.
"Generalized exponentially (s, m)–convex functions and associated algebraic properties"
1) Put comma at the end of every inequality.
2) In the proof of Proposition 2, what is the meaning of “b” above line 128.
3) Page 6, in first line, what is the meaning of “b”, remove the type error.
4) In the proof of theorem 4, what is the meaning of “b” and “a” above line 141.
We have rewritten all ot them.
Hermite-Hadamard type inequality via Generalized exponentially (s, m)–convex functions
We modified the introduction to provide the motivation for the study, please see the text.
2) Statement of the Corollary 2.11 and 2.15 are replace by “ If s = 1, then we obtain the following inequality”.
We have rewritten it.
3) Statement of the Corollary 2.12 and 2.16 are replace by “ If m = 1, then we obtain the following inequality”.
We have rewritten it.
4) Statement of the Corollary 2.13 and 2.17 are replace by “ If s = m = 1, then we obtain the following inequality”.
We have rewritten it.
5) For the interest of readers and quality of work, must add some examples in the sense of symmetry using the above theorems.
We have modified the manuscript and we have added some examples and applications please the full text. Regarding the limitation we may refer to the remark 1 that the present study is extension and generalization of many well-known works,
Applications
- Application section in the current form is not enough. Must Revise.
- Add some applications related q-digamma and quadrature type.
We have added
Dear Professor, Thank again for the very useful comments and we have modified the manuscript under your suggestion as much as possible and looking forward your kind further suggestions if there are some more.
Reviewer 3 Report
The author has improved the content of the paper. I think the paper meets now the standards for publication.
Author Response
Dear Professor, we really appreciate your suggestions and comments and we are very grateful for your recommendation.